# Enhancing Zinc Biofortification of Wheat through Integration of Zinc, Compost, and Zinc-Solubilizing Bacteria

Shah Khalid [1], Amanullah [1,]*[ ] and Iftikhar Ahmed [2]

1    Department of Agronomy, The University of Agriculture, Peshawar 25130, Pakistan;
     khalidmashaal@aup.edu.pk
2    National Microbial Culture Collection of Pakistan (NCCP), Bioresource Conservation Institute (BCI),
     National Agricultural Research Center (NARC), Islamabad 45500, Pakistan; iftikhar.ahmed@parc.gov.pk
*    Correspondence: amanullah@aup.edu.pk

**Abstract:** Zinc (Zn) deficiency is a fairly widespread agronomic constraint in many of the world's cereal (wheat, rice, corn, barley, etc.) production regions. Zinc is an imperative micronutrient required for optimum plant growth and development. Low Zn availability in about 50% of global land has resulted in Zn deficiency in cereal grains. A two-year field experiment was conducted at the Agronomy Research Farm, The University of Agriculture, Peshawar, during Rabi season 2018–19 (Y1) and 2019–20 (Y2) to study the impact of Zn levels (0, 5, 10 and 15 kg Zn ha$^{-1}$), compost types (control, composted sheep manure (SMC), composted poultry manure (PMC) and farmyard manure compost (FYMC), and Zn-solubilizing bacteria (ZnSB) (with (+) and without (-) on Zn biofortification in order to overcome Zn deficiency. The experiment was set up in three replications in a randomized complete block design. The wheat variety "Pirsabak-2013" was planted in a 30 cm row-to-row spacing. The plot size was kept at 9 cm$^2$, with 10 rows plot$^{-1}$, and the seed was sown at a rate of 100 kg ha$^{-1}$. The results showed that ZnSB application increased ShZnC (shoot Zn concentration) to a maximum level of 29.3 mg kg$^{-1}$, ShZnUp (shoot Zn uptake) to 176.0 g ha$^{-1}$, SZnUp (straw Zn uptake) to 116.67 g ha$^{-1}$, and TZnUp (total Zn uptake) to 230.3 g ha$^{-1}$. In the case of compost types, PMC resulted in maximum grain Zn uptake (GZnUp) (28.9 mg kg$^{-1}$), ShZnUp (192.9 g ha$^{-1}$), GZnC (33.4 mg kg$^{-1}$), GZnUp (125.06 g ha$^{-1}$), SZnUp (125.26 g ha$^{-1}$), and TZnUp (250.3 g ha$^{-1}$). In the case of Zn levels, higher ShZnC (31.5 mg kg$^{-1}$), ShZnUp (191.3 g ha$^{-1}$), GZnC (34.4 mg kg$^{-1}$), SZnC (23.5 mg kg$^{-1}$), GZnUp (128.98 g ha$^{-1}$), SZnUp (129.29 g ha$^{-1}$), and TZnUp (258.3 g ha$^{-1}$) were calculated with the use of the highest rate of 15 kg Zn ha$^{-1}$, which was either statistically similar to or followed by 10 kg Zn ha$^{-1}$. A strong positive correlation was found among uptake by different plant parts (ZnG, ZnS, ShZnUp, GZnUp, SZnUp, and TZnUp). It was concluded that the combined application of PMC and 10 kg Zn ha$^{-1}$ along with ZnSB (+) improved Zn biofortification and uptake in wheat crop under Zn-deficient soils.

**Keywords:** wheat; biofortification; organic sources; Zn levels; zinc-solubilizing bacteria

## 1. Introduction

Micronutrient deficiency, especially of zinc (Zn), is a common problem in developing countries [1–3]. Chronic low Zn consumption, especially from cereal-based diets, is the leading cause of Zn deficiency, and biofortification is a promising technique for addressing this problem by boosting Zn concentrations in edible crops [4,5]. Wheat (*Triticum aestivum* L.), which is a major staple food crop around the world, has naturally low Zn contents in its grains [6–8]. Grain Zn in wheat produced in key producing regions currently ranges from 20 to 30 mg kg$^{-1}$, with an average of 27.3 mg kg$^{-1}$ on a global scale, leaving a large gap between the biofortification target for human health and the current situation [4,9,10].

Zn deficiency affects plant growth and crop yields. Zn shortage in the soil may be mirrored in plants, posing major health concerns to humans and animals through

plant-based meals [9,11]. Zn plays a significant role in root growth [12], increased crop yield [9,13–15], plant resistance against diseases, photosynthesis, cell membrane integrity, pollen formation, energy production [16,17], and the enhancement of antioxidant enzymes and chlorophyll within plant tissues [18]. Furthermore, Zn is essential for the production of phytohormones such as abscisic acid, auxin, gibberellins, and cytokinin, and its shortage impairs plant cell proliferation. In addition, Zn is required for the activity of over 300 enzymes, including transphosphorylases, aldolases, dehydrogenases, isomerases, DNA, and RNA polymerases [19]. On the other hand, low Zn-content plants are more susceptible to fungal infections, as well as photo- and heat damage [20].

Due to high soil pH, low organic matter, and low soil moisture, limited Zn availability contributes to Zn insufficiency in agricultural soils under arid and semi-arid climates. Pakistani soils have a high pH, low organic matter, and low moisture content, all of which influence Zn uptake from the soil rhizosphere. Pakistan's soils are naturally alkaline [21], and crops grown on calcareous soils are particularly susceptible to zinc deficiency [20–22]. Additionally, Zn insufficiency is prevalent in wheat-growing nations [20]. Nearly 70% of Pakistan's agricultural land is lacking in phytoavailable zinc. Within seven days of application, approximately 96 to 99 percent of inorganic applied Zn is transformed into various insoluble forms, depending on the soil type and physicochemical interactions of the soil [23–25].

Soil Zn is found in solution, exchangeable form, and associated with soil organic matter [26]. Plants can take up Zn as a divalent cation ($Zn^{2+}$), but only a small amount of total Zn is soluble in soil solution (>1 mg Zn $kg^{-1}$) due to complexation, adsorption, and precipitation [27]. Soil pH [28,29], clay type [30], total carbonate [31,32], and Zn applied [33] all affect the adsorption-desorption process and soil Zn solubility [34,35]. Organic matter is commonly used as a chelating agent, reducing Zn adsorption and increasing the production of soluble organic-Zn complexes, increasing Zn availability [36].

There are various ways to avoid zinc deficiency in edible crop components. Genetic biofortification requires long-term breeding and biotechnological technologies that may not work in all soil environments [9]. Thus, agronomic biofortification strategies such as zinc fertilization, organic manure application [37] and microbial inoculants, i.e., zinc-solubilizing bacteria (ZnSB), are gaining importance [38].

Zinc-solubilizing bacteria may be used in place of Zn supplementation because they convert applied inorganic Zn to usable forms. Certain rhizobacteria species are capable of mobilising Zn from soils in an accessible form. Bacteria capable of solubilizing Zn compounds do so by producing and excreting organic acids [39]. These bacteria promote plant growth and development by dissolving complicated Zn compounds into simpler ones, thus increasing the amount of Zn available to the plants. Microorganisms that solubilize Zn do so by a variety of processes, one of which is acidification [40]. These microorganisms create organic acids in the soil, which act as a sink for Zn cations and lower the pH of the surrounding soil. Additional methods for Zn solubilization may include the formation of siderophores and protons, oxidoreductive systems on cell membranes, and chelated ligands [41]. Researchers have also evaluated potential Zn-solubilizing bacteria for improved nutrition and Zn uptake, Zn-solubilizing bacterial strains that modify growth and yield, and Zn-biofortified wheat [42]. Vaid et al. [43] observed that inoculating rice with Zn-solubilizing bacteria promoted rice growth and enhanced Zn nutrition by 42.7 percent. As a result, it is critical to biofortify wheat with Zn to reduce this risk of Zn deficiency in humans [9]. Two important options for Zn biofortification of food crops are agronomic interventions, which include proper fertilizer use, and genetic improvements, which include plant breeding and genetic modification [5,9].

The current study was undertaken to investigate the role of Zn-solubilizing bacteria, Zn levels, and composted organic manures on Zn availability to plants for enhancing Zn uptake in wheat, keeping in mind the need for an integrated strategy for zinc biofortification.

## 2. Materials and Methods

During the Rabi (winter) seasons of 2018–19 (Y1) and 2019–20 (Y2), a field experiment was conducted at The University of Agriculture, Peshawar's Agronomy Research Farm to investigate the impact of Zn levels, compost types, and Zn-solubilizing bacteria on wheat productivity, Zn biofortification, and zinc use efficiency. The experiment was designed using three replications in a randomized complete block design (RCBD). The wheat variety "Pirsabak-2013" was planted at a row-to-row spacing of 30 cm. The plot size was kept at 9 m$^2$ (3 m width × 3 m length), with 10 rows per plot and a row length of 3 m. The soil was ploughed twice with a cultivator, then with a rotavator to break up clods and pulverize them. In each subplot, the seed was sown at a rate of 120 kg ha$^{-1}$ using a hand hoe. All other production procedures, including planting, irrigation, weeding, hoeing, harvesting, and threshing/shelling, were performed consistently across all treatments. Before beginning the experiment, a composite soil sample was collected from the field and evaluated for its physiochemical parameters (Table 1). The mean maximum and minimum temperatures, as well as the mean monthly rainfall, were measured for the two wheat-growing seasons (Figure 1).

**Table 1.** Pre-sowing soil properties of experimental site.

| Soil Properties | Values |
| --- | --- |
| AB-DTPA Zn (mg kg$^{-1}$) | 0.67 |
| CaCO$_3$ (%) | 16.6 |
| Clay (%) | 11 |
| ECe (dS m$^{-1}$) | 0.86 |
| Organic matter (%) | 0.68% |
| pH | 7.99 |
| Sand (%) | 36 |
| Silt (%) | 53 |
| Texture | Silty clay loam |

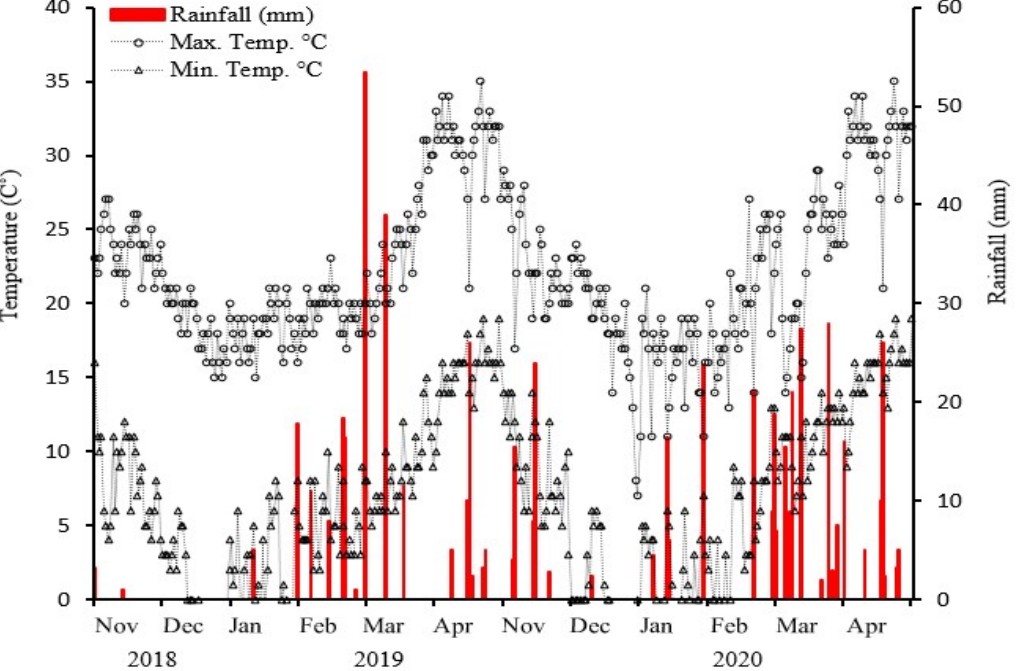

**Figure 1.** Daily maximum and minimum temperature (°C) and rainfall (mm) of the experimental site during 2018–2019 to 2019–2020.

### 2.1. Experimental Site

The experimental site is located at 34° N latitude and 71° E longitude, at an elevation of 358 m and 1600 km north of the Arabian Sea. The experimental site's climate is hot, semi-arid, subtropical, and continental, with an average rainfall of 403 mm. Furthermore, the average maximum temperature in the summer (from May to September) is 40 °C, and the average lowest is 25 °C, while the average maximum temperature in the winter (from December to March) is 18.4 °C, and the average minimum temperature is 4 °C. Irrigation is provided by the Warsak canal of the Kabul River. The soil texture is clay loam with 0.87 percent organic matter, exchangeable K (121 mg kg$^{-1}$), extractible P (5.57 mg kg$^{-1}$) and an alkaline pH of 8.2 with 0.73 Zn mg kg$^{-1}$ (Table 1). In this experiment, the following levels of factors used were: zinc-solubilizing bacteria (with and without), zinc levels (0, 5, 10, and 15 kg Zn ha$^{-1}$), and compost types (composted FYM, sheep manure and poultry manure). Nitrogen was administered to the experimental plots at a rate of 120 kg N ha$^{-1}$ and phosphorus at a rate of 90 kg $P_2O_5$ ha$^{-1}$ in the form of DAP (46 percent $P_2O_5$ and 18 percent N), with the remaining nitrogen compensated from urea (46 percent N). All phosphorus was applied at sowing time, whereas nitrogen was applied in three equal splits, namely at sowing time, after the first irrigation (21 days after sowing), and after the second irrigation (45 days after sowing), according to the recommended irrigation schedule. The chemical herbicides Buctril Super and Puma Super were applied at rates of 1.5 kg ha$^{-1}$ and 1.25 lit ha$^{-1}$, respectively, for weed control of board leaf and grassland weeds after the first irrigation, when the majority of the weeds had emerged.

### 2.2. Factors in Detail

Zinc-solubilizing bacterial strains were taken from the Institute for Microbial Culture Collection of Pakistan, NARC, Islamabad, in cultured form, isolated from different rhizospheres. In this research, a combination of five different bacterial strains, namely *Alcaligenes pakistanensis*, *Sphingobacterium pakistanensis*, *Cellolumonas pakistanensis*, *Pseudomonas* sp. and *Bacillus* sp., was used. Zinc-solubilizing bacteria were applied to the seed before sowing. A specific amount of solution was maintained for the seed of each treated plot containing $10^8$ mL$^{-1}$ active cells of the bacteria, which soaked the seeds thoroughly. Zinc sulphate was used as a source for zinc in three levels (5, 10, 15 kg Zn ha$^{-1}$) along with one control (0 kg Zn ha$^{-1}$). Zinc was applied during sowing time to the soil. Three different types of composts (poultry, cattle and sheep manure) were applied at a rate of 5 t ha$^{-1}$ during sowing, along with one control (no compost application). Cattle and sheep manures were taken from the Dairy Farm, and poultry manure was taken in the prescribed quantity from Poultry Farm of The Agriculture University, Peshawar and was kept in pits to rot for 45 days. A pre-sowing chemical analysis was carried out for the three compost types (Table 2).

**Table 2.** Chemical composition of different composts used in the study.

| Nutrients | Poultry Manure Compost | Sheep Manure Compost | Farmyard Manure Compost |
|---|---|---|---|
| N (%) | 1.5 | 1.2 | 1.10 |
| P (%) | 1.15 | 1.03 | 1.01 |
| K (%) | 1.7 | 1.63 | 2.86 |
| AB-DTPA Zn (mg kg$^{-1}$) | 156 | 134 | 72 |
| AB-DTPA Fe (mg kg$^{-1}$) | 557 | 423 | 296 |

### 2.3. Soil and Plant Zinc Analysis

Zn levels in wheat plants (grain and straw) were measured after wheat crop harvesting in each treatment using the conventional procedures described by [44]. After harvest, grains were sampled at random from all treatments to determine their Zn content. A sample of 0.5 g dry plant material (straw and grain) was placed in a flask and treated with 10 mL pure $HNO_3$ for 24 h before being roasted on a hot plate until white fumes appeared. After the white fumes dissipated, the samples were removed from the hot plate, and the digest

was diluted with 100 mL of water before being filtered using Whatman-42 filter paper in a 100 mL volumetric flask. The Zn concentration in the filtrate was determined using an atomic absorption spectrophotometer [44]. The following formula was used to calculate the Zn concentration in grains and straw.

$$\text{Shoot or Grain or straw Zn conc. } (\text{mg kg}^{-1}) = \frac{\text{Machine Reading}}{\text{Weight of Grain or straw}} \times \text{volume of extract (mL)} \tag{1}$$

$$\text{Total Zn uptake } (\text{g ha}^{-1}) = (\text{Zn Uptake in grain}) + (\text{Zn Uptake in straw}) \tag{2}$$

$$\text{Zn uptake of straw } (\text{g ha}^{-1}) = \frac{\text{Straw Zn conc. } \left(\text{mg kg}^{-1}\right) \times \text{straw yield } \left(\text{kg ha}^{-1}\right)}{1000} \tag{3}$$

$$\text{Zn uptake of grain } (\text{g ha}^{-1}) = \frac{\text{Zn conc. of grain } \left(\text{mg kg}^{-1}\right) \times \text{grain yield } \left(\text{kg ha}^{-1}\right)}{1000} \tag{4}$$

### 2.4. Statistical Analysis

The data were statistically analyzed by analysis of variance (ANOVA) using the technique given by Gomez and Gomez (1984). The Origin and Microsoft Excel programmes were utilized for statistical analysis. The LSD test was used to examine the significance of differences in mean values at $p = 0.05$ (Steel and Torrie, 1980).

## 3. Results

### 3.1. Shoot Zinc Concentrations (mg kg$^{-1}$)

The interactive effects of ZnSB $\times$ C, Zn $\times$ C, and Y $\times$ ZnSB $\times$ C significantly affected shoot zinc concentrations (ShZnC) of wheat, while the remaining interactions had no significant effect on ShZnC (Tables 3 and 4). The application of ZnSb increased the ShZnC of wheat by 7.5% over no application of ZnSB. PMC enhanced ShZnC (16%) over control (no compost application); this was followed by SMC (12.8%). The application of Zn at the rate of 15 kg Zn ha$^{-1}$ increased ShZnC (31.5%) over control (no Zn application); this was followed by 10 kg Zn ha$^{-1}$ (29.7%) and 5 kg Zn ha$^{-1}$ (26.1%). Higher ShZnUp (29.0 g ha$^{-1}$) was recorded during 2019–20 than in 2018–19 (27.3 mg kg$^{-1}$). The interactive effect of ZnSB $\times$ C indicated that ZnSB along with compost significantly improved wheat ShZnC, and PMC produced higher ShZnC with ZnSB application (Figure 2A). The interaction of Zn $\times$ C showed that among different types of compost, PMC increased ShZnC with 15 kg Zn ha$^{-1}$ (Figure 2B). The interaction of the application of Y $\times$ ZnSB $\times$ C showed that during 2018–19, PMC produced higher ShZnC during both years both with and without ZnSB application; however, during 2019–20, SMC achieved higher values of ShZnC with ZnSB application (Figure 2C). ShZnC showed to be significantly positively correlated with shoot zinc uptake, grain zinc concentration, straw zinc concentration, grain zinc uptake, straw zinc uptake, and total zinc uptake (Table 5).

**Table 3.** Mean square of shoot zinc (Zn) concentration (mg kg$^{-1}$) of wheat as affected by zinc, compost and zinc-solubilizing bacteria.

| SOV | DF | ShZnC | ShZnUp | GZnC | GZnUp | SZnC | SZnUp | TZnUp |
|---|---|---|---|---|---|---|---|---|
| Years (Y) | 1 | 140.53 | 21,205.62 | 152.12 | 8521.96 | 178.5 | 11,282.5 | 11,282.50 |
| Reps within year | 4 | 11.25 | 611.50 | 0.81 | 85.98 | 2.2 | 89.9 | 89.90 |
| Zinc-solubilizing bacteria (ZnSB) | 1 | 236.25 | 22,692.20 | 155.24 | 6802.19 | 193.5 | 12,795.7 | 12,795.66 |
| Compost types (C) | 3 | 205.76 | 35,710.52 | 770.23 | 20,956.48 | 282.5 | 18,534.3 | 18,534.29 |
| Zinc levels (Zn) | 3 | 997.86 | 53,541.10 | 1665.36 | 41,167.62 | 943.5 | 33,580.3 | 33,580.31 |
| ZnSB $\times$ C | 3 | 5.92 | 924.36 | 3.79 | 276.52 | 16.1 | 94.5 | 94.54 |
| ZnSB $\times$ Zn | 3 | 1.38 | 179.44 | 9.40 | 328.74 | 9.9 | 527.0 | 526.97 |
| Zn $\times$ C | 9 | 16.32 | 1202.34 | 25.54 | 867.67 | 26.8 | 981.0 | 981.03 |
| Y $\times$ ZnSB | 1 | 0.01 | 551.53 | 0.64 | 7.47 | 45.6 | 65.2 | 65.24 |

**Table 3.** *Cont.*

| SOV | DF | ShZnC | ShZnUp | GZnC | GZnUp | SZnC | SZnUp | TZnUp |
|---|---|---|---|---|---|---|---|---|
| Y × C | 3 | 2.30 | 514.60 | 21.14 | 1107.48 | 18.0 | 455.6 | 455.59 |
| Y × Zn | 3 | 2.35 | 312.58 | 8.91 | 570.96 | 10.6 | 556.2 | 556.22 |
| Zn × ZnSB × C | 9 | 1.69 | 79.23 | 1.14 | 141.49 | 2.0 | 76.9 | 76.87 |
| Y × ZnSB × C | 3 | 4.66 | 2044.57 | 15.68 | 263.73 | 6.0 | 1016.0 | 1016.03 |
| Y × ZnSB × Zn | 3 | 0.38 | 59.55 | 4.67 | 192.01 | 0.2 | 18.3 | 18.29 |
| Y × Zn × C | 9 | 1.13 | 71.54 | 3.23 | 161.04 | 2.6 | 66.9 | 66.94 |
| Y × Zn × ZnSB × C | 9 | 1.75 | 372.52 | 3.24 | 85.03 | 6.3 | 274.7 | 274.70 |
| Error | 124 | 1.48 | 77.58 | 1.23 | 60.96 | 1.0 | 81.8 | 81.81 |
| Total | 191 | | | | | | | |

**Table 4.** ShZnC, ShZnUp, GZnC, SZnC, GZnUp, SZnUp and TZnUp of wheat as affected by zinc, compost, and zinc-solubilizing bacteria.

| Zinc Solubilizing Bacteria (ZnSB) | ShZnC | ShZnUp | GZnC | GZnUp | SZnC | SZnUp | TZnUp |
|---|---|---|---|---|---|---|---|
| Without ZnSB | 27.1 b | 154.2 a | 29.7 b | 101.75 b | 19.0 b | 100.34 b | 202.1 b |
| With ZnSB | 29.3 a | 176 b | 31.5 a | 113.66 a | 21.0 a | 116.67 a | 230.3 a |
| LSD$_{(0.05)}$ for ZnSB | 0.3 | 2.5 | 0.3 | 2.23 | 0.3 | 2.58 | 2.6 |
| Compost types (C) | | | | | | | |
| Control (no compost) | 25.2 d | 128.5 d | 24.6 d | 78.53 d | 16.5 d | 80.95 d | 159.5 d |
| Composted farmyard manure | 28.5 c | 163.2 c | 31.8 c | 107.00 c | 20.5 c | 108.42 c | 215.4 c |
| Composted sheep manure | 28.9 b | 175.9 b | 32.4 b | 120.23 b | 21.8 a | 119.40 b | 239.6 b |
| Composted poultry manure | 30.0 a | 192.9 a | 33.4 a | 125.06 a | 21.4 b | 125.26 a | 250.3 a |
| LSD$_{(0.05)}$ for compost types | 0.5 | 3.6 | 0.4 | 3.15 | 0.4 | 3.65 | 3.7 |
| Zinc levels (kg ha$^{-1}$) | | | | | | | |
| 0 | 21.5 d | 117.3 d | 21.9 d | 65.41 d | 13.6 d | 70.29 d | 135.7 d |
| 5 | 29.1 c | 167.6 c | 32.2 c | 110.79 c | 20.8 c | 112.00 c | 222.8 c |
| 10 | 30.6 b | 184.3 b | 33.9 b | 125.63 b | 22.3 b | 122.45 b | 248.1 b |
| 15 | 31.5 a | 191.3 a | 34.4 a | 128.98 a | 23.5 a | 129.29 a | 258.3 a |
| LSD$_{(0.05)}$ for zinc levels | 0.5 | 3.6 | 0.4 | 3.15 | 0.4 | 3.65 | 3.7 |
| Year | | | | | | | |
| 2018–19 | 27.3 b | 154.3 b | 29.7 b | 101.04 | 19.1 b | 100.84 b | 201.9 b |
| 2019–20 | 29.0 a | 175.6 a | 31.5 a | 114.37 | 21.0 a | 116.17 a | 230.5 a |
| Significance | * | ** | ** | ** | ** | ** | ** |
| Interaction | | | | | | | |
| ZnSB × C | Figure 2A | Figure 2D | Figure 4A | ns | Figure 8A | ns | Figure 11A |
| ZnSB × Zn | ns | ns | Figure 4B | Figure 6A | Figure 8B | Figure 9D | Figure 11B |
| Zn × C | Figure 2B | Figure 3A | Figure 4C | Figure 6B | Figure 8C | Figure 10A | Figure 11C |
| Y × ZnSB | ns | Figure 3B | ns | ns | Figure 8D | ns | ns |
| Y × C | ns | Figure 3C | Figure 4D | Figure 6C | ns | Figure 10B | Figure 11D |
| Y × Zn | ns | ns | Figure 5A | Figure 6D | ns | Figure 10C | Figure 12A |
| Zn × ZnSB × C | ns | ns | ns | Figure 7A | Figure 9A | ns | Figure 12B |
| Y × ZnSB × C | Figure 2C | Figure 3D | Figure 5B | Figure 7B | Figure 9B | Figure 10D | Figure 12C |
| Y × ZnSB × Zn | ns | ns | Figure 5C | Figure 7C | ns | ns | ns |
| Y × Zn × C | ns | ns | Figure 5D | Figure 7D | Figure 9C | ns | Figure 12D |
| Y × Zn × ZnSB × C | ns | ns | ns | ns | ns | ns | ns |

Note: ShZnC, ShZnUp, GZnC, SZnC, GZnUp, SZnUp and TZnUp stand for shoot zinc concentration, shoot zinc uptake, grain zinc concentration, straw zinc concentration, grain zinc uptake, straw zinc uptake and total zinc uptake. *, ** indicate that data are significant at 5% and 1% level of probability, respectively, and ns stand for non-significant data at the 5% level of probability. Different letters represent significant differences ($p < 0.05$).

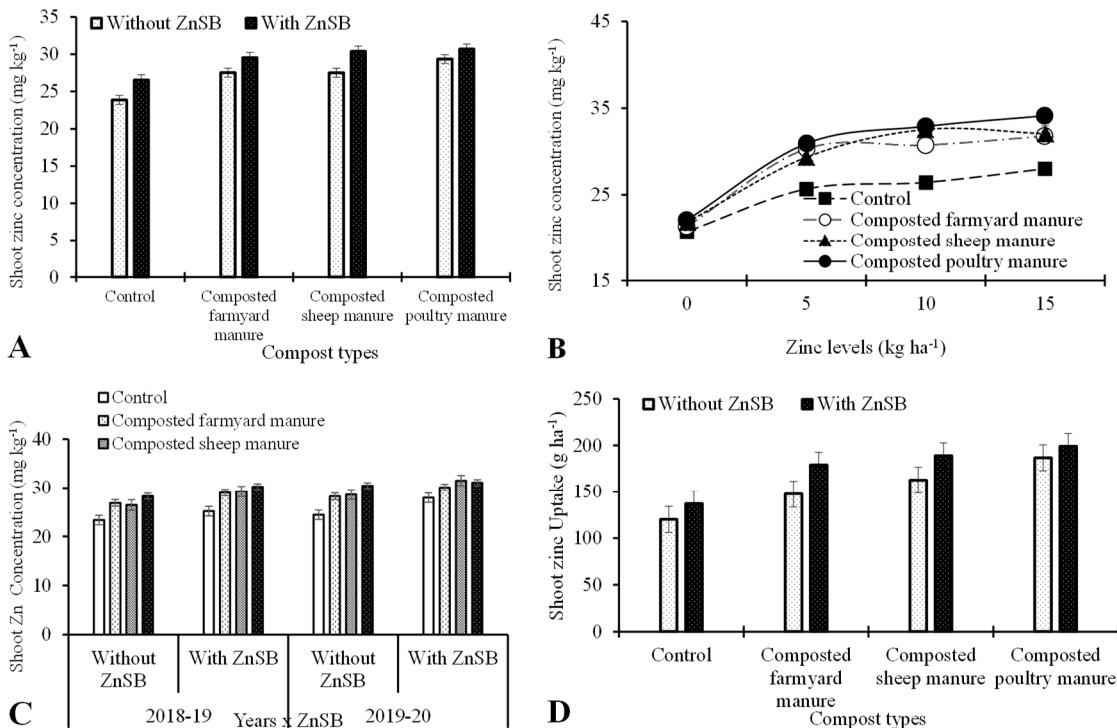

**Figure 2.** Interactive effect of compost types and ZnSB (**A**), compost types and Zn levels (**B**), years, ZnSB and compost types on shoot Zn concentrations of wheat (**C**) and interactive effect of compost types and ZnSB on shoot Zn uptake of wheat (**D**).

**Table 5.** Correlation analysis of yield and yield components, zinc concentration and uptake with different parameters of wheat (Note: * indicate that data are significant at 5% level of probability).

|  | ShZnC | ShZnUp | GZnC | GZnUp | SZnC | SZnUp | TZnUp |
|---|---|---|---|---|---|---|---|
| ShZnC | 1 | 0.95736 * | 0.97247 * | 0.97414 * | 0.96452 * | 0.96804 * | 0.97027 * |
| ShZnUp | 0.95736 * | 1 | 0.95697 * | 0.94056 * | 0.98415 * | 0.98195 * | 0.98702 * |
| GZnC | 0.97247 * | 0.95697 * | 1 | 0.97209 * | 0.97263 * | 0.97755 * | 0.97914 * |
| GZnUp | 0.97414 * | 0.94056 * | 0.97209 * | 1 | 0.97524 * | 0.97413 * | 0.97863 * |
| SZnC | 0.96452 * | 0.98415 * | 0.97263 * | 0.97524 * | 1 | 0.9838 * | 0.99569 * |
| SZnUp | 0.96804 * | 0.98195 * | 0.97755 * | 0.97413 * | 0.9838 * | 1 | 0.99619 * |
| TZnUp | 0.97027 * | 0.98702 * | 0.97914 * | 0.97863 * | 0.99569 * | 0.99619 * | 1 |

*3.2. Shoot Zn Uptake (g ha$^{-1}$)*

ZnSB application enhanced ShZnUp by 12.3% over no application of ZnSB. Among compost types, PMC increased ShZnUp by 33.3% over control, which was followed by SMC (26.9%) and FYMC (2.5%). The ShZnUp of wheat showed a positive response toward Zn application, recording a 38.6% increase in the ShZnUp of wheat with 15 kg Zn ha$^{-1}$, followed by 10 kg Zn ha$^{-1}$ (36.6%) and 5 kg Zn ha$^{-1}$ (30%), in comparison with control. Higher ShZnUp (175.6 g ha$^{-1}$) was recorded during 2019–20 than in 2018–19 (154.3 g ha$^{-1}$). The interactive effects of ZnSB × C, Zn × C, Y × ZnSB, Y × C, Y × Zn and Y × ZnSB × C significantly affected ShZnUp, while the rest of the interactions did not significantly affect the ShZnUp of wheat (Tables 3 and 4). The interactive effect of ZnSB × C indicated that the application of ZnSB and compost significantly improved wheat ShZnUp, and PMC produced higher ShZnUp with the application of ZnSB (Figure 2D). The Zn × C interaction showed that among different types of compost, PMC increased ShZnC with each increment of Zn (Figure 3A). The interaction of Y × ZnSB showed that during each year, ZnSB application

produced higher ShZnUp; however, higher values of ShZnUp were recorded during 2019–20 as compared with 2018–19 (Figure 3B). The interaction of Y × C revealed that higher ShZnUp was recorded with the application of PMC during both years; however, higher values of ShZnUp were recorded during 2019–20 as compared with 2018–19 (Figure 3C). Additionally, the interaction of Y × ZnSB × C showed that PMC produced higher ShZnUp with the application of ZnSB during both years; however, during 2019–20, higher values of ShZnUp were recorded as compared with 2018–19 (Figure 3D). The correlation table shows that shoot zinc uptake was significantly positively correlated with shoot Zn concentration, grain zinc concentration, straw zinc concentration, grain zinc uptake, straw zinc uptake, and total zinc uptake (Table 5).

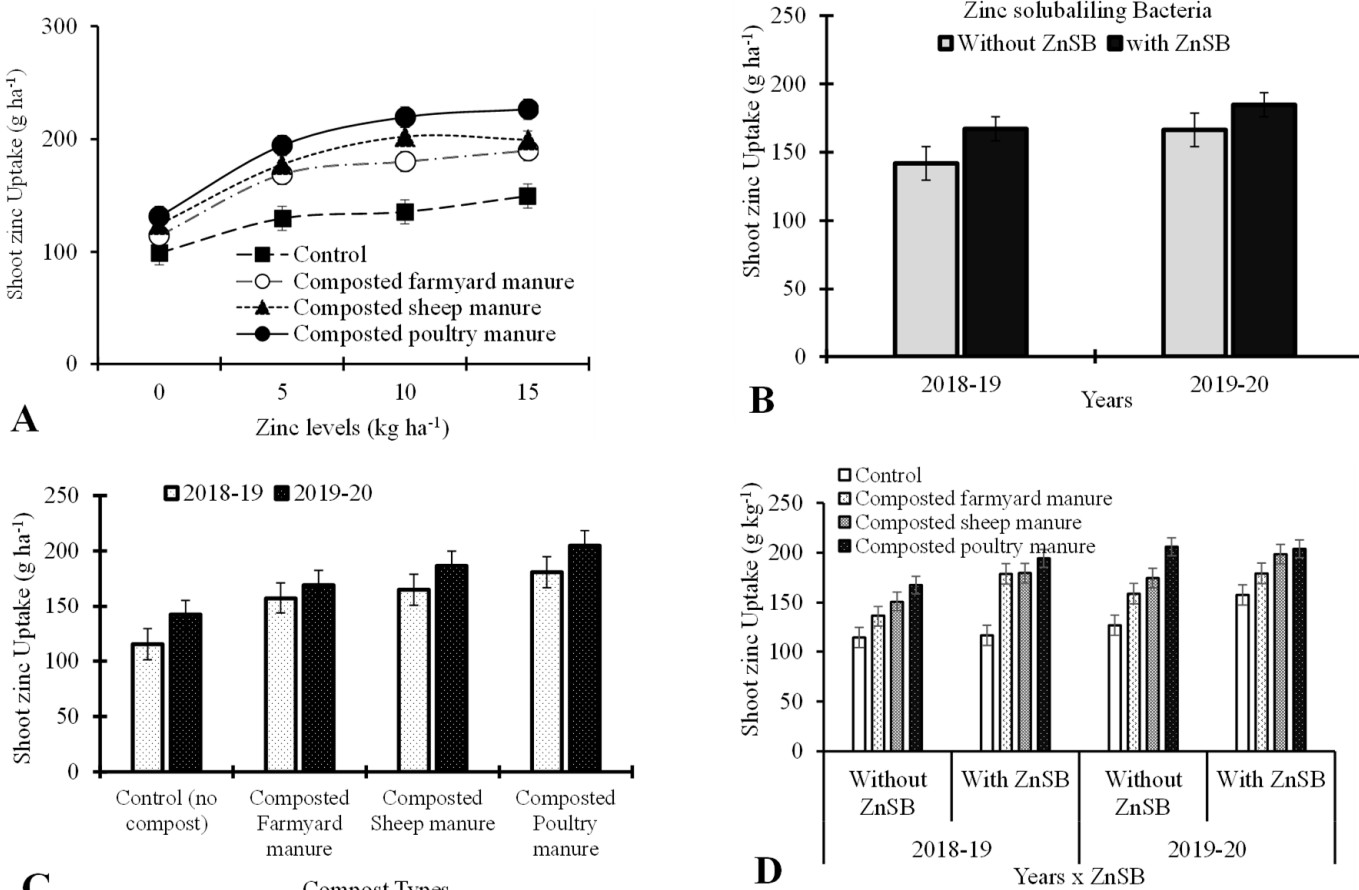

**Figure 3.** Interactive effect of Zn levels and compost types (**A**), years and ZnSB (**B**), compost types and ZnSB (**C**), and years, ZnSB and compost types on shoot Zn uptake of wheat (**D**).

### 3.3. Grain Zinc Concentrations (mg kg$^{-1}$)

Grain zinc concentration (GZnC) was enhanced by 5.7% with the application of ZnSB (+) compared to without ZnSB (-) application. The application of PMC significantly increased the GZnC of wheat by 26.3% over the control, which was followed by SMC (24%), as shown in Table 4. In the case of Zn levels, GZnC increased by 36.3% over the control with the application of Zn at the rate of 15 kg ha$^{-1}$, followed by 10 kg Zn ha$^{-1}$ (35.3%). Higher values of GZnC (31.5 mg kg$^{-1}$) were recorded during 2019–20 than in 2018–19 (29.7 mg kg$^{-1}$). The interactive effect of ZnSB × C indicated that the application of ZnSB with PMC enhanced GZnC (Figure 4A). The interaction of ZnSB × Zn indicated that increasing Zn levels from 0 to 10 kg Zn ha$^{-1}$ increased GZnC with ZnSB application, while further increases in Zn level did not considerably increase the GZnC of wheat (Figure 4B). The interaction of Zn × C indicated that among different types of compost, poultry manure and sheep compost increased GZnC with each increment in Zn up to 10 kg Zn ha$^{-1}$.

Further increases in Zn level slightly decreased the value of GZnC with the application of SMC (Figure 4C). The interaction of Y × C revealed that higher GZnC was recorded with the application of PMC during both years; however, during 2019–20, SMC produced values statistically similar to poultry manure (Figure 4D). The interaction of Y × Zn indicated that higher values of GZnC were recorded with the application of 15 kg Zn ha$^{-1}$ during both years; however, during 2018–19, the application of 10 kg Zn ha$^{-1}$ produced values statistically similar to 15 kg Zn ha$^{-1}$ (Figure 5A). Additionally, the interaction of Y × ZnSB × C showed that PMC and SMC produced higher values of GZnC with the application of ZnSB during both years; however, during 2019–20, higher values of GZnC were recorded as compared with 2018–19 (Figure 5B). Similarly, the interactive effect of Y × ZnSB × Zn showed that 10 kg Zn ha$^{-1}$ produced higher GZnC with the application of ZnSB during both years. Further increases in Zn level considerably increased GZnC (Figure 5C). Likewise, the interactive effect of Y × Zn × C showed that all types of composts, particularly PMC and SMC, produced higher values of GZnC with each increment of Zn up to 10 kg Zn ha$^{-1}$ during both years; however, further increases in Zn level considerably increased GZnC (Figure 5D). The correlation table shows that grain zinc concentration was significantly positively correlated with shoot Zn concentration, shoot zinc uptake, straw zinc concentration, grain zinc uptake, straw zinc uptake, and total zinc uptake (Table 5).

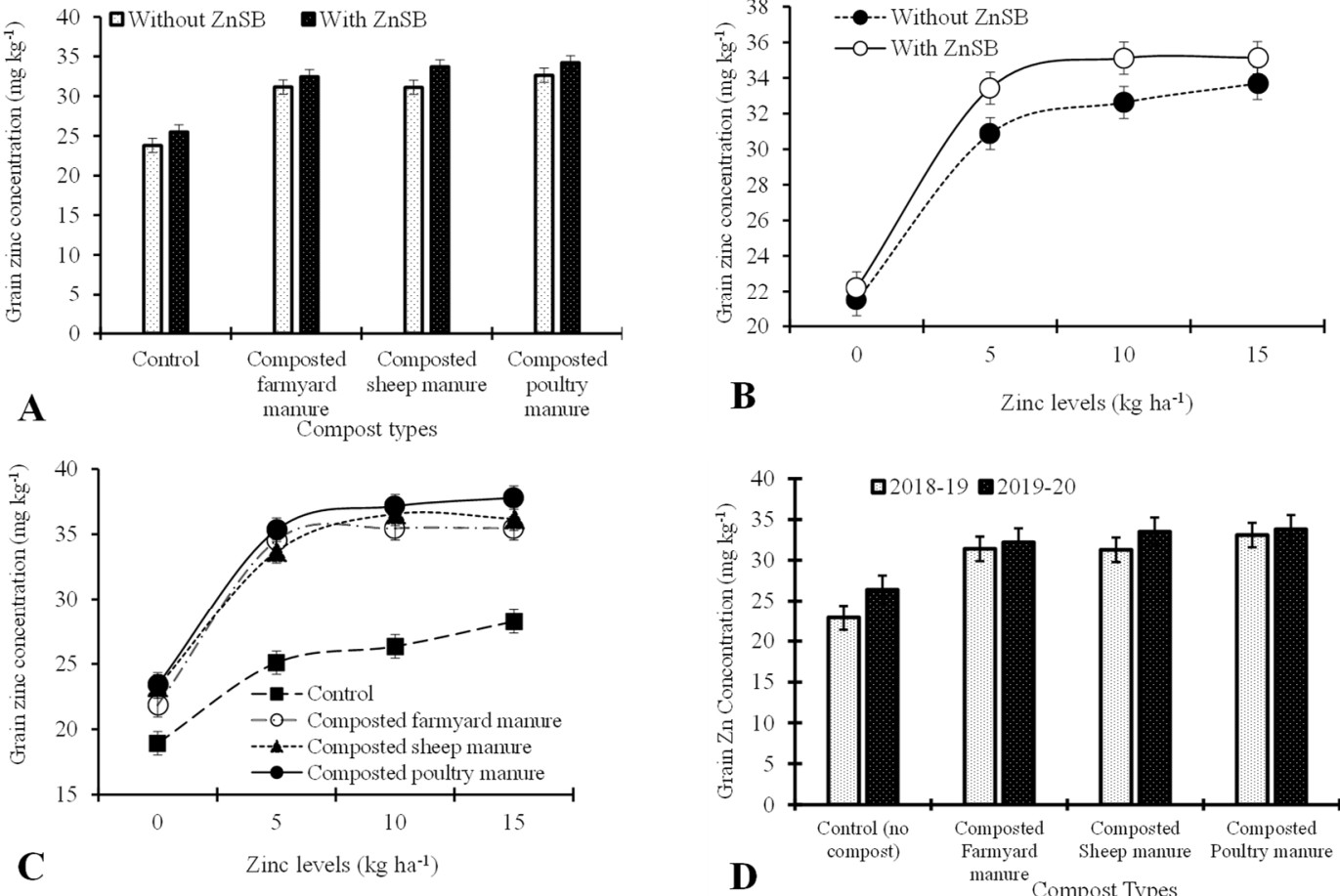

**Figure 4.** Interactive effect of ZnSB and compost types (**A**), ZnSB and Zn levels (**B**), Zn levels and compost types (**C**), and compost types and years on grain Zn concentration of wheat (**D**).

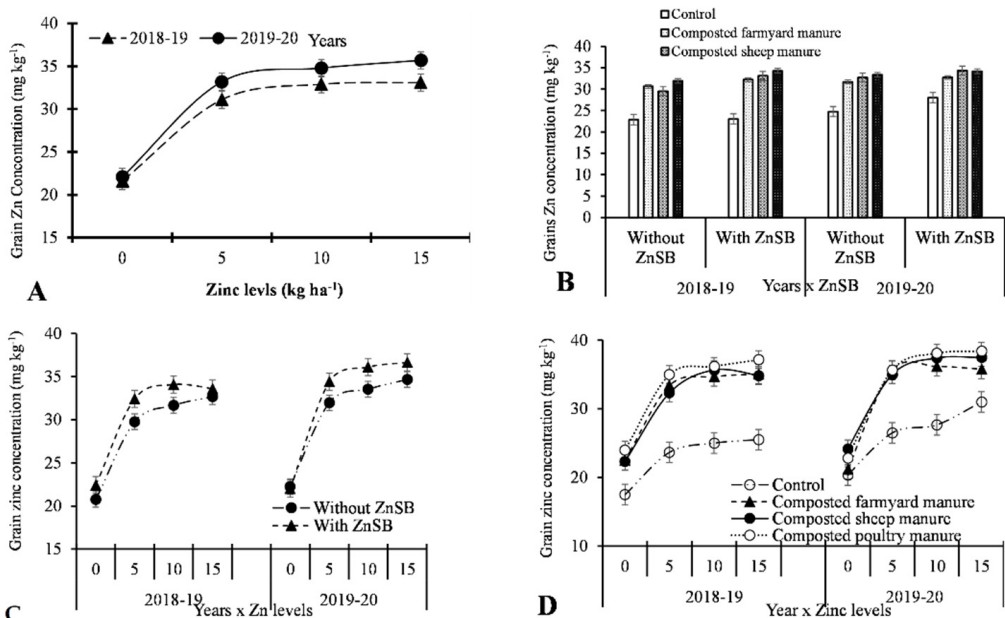

**Figure 5.** Interactive effect of Zn levels and years (**A**), years, ZnSB and compost types (**B**), years, Zn levels and ZnSB (**C**), and years, Zn levels and compost types on grain Zn concentration of wheat (**D**).

### 3.4. Grain Zinc Uptake ($g\ ha^{-1}$)

ZnSB enhanced grain zinc uptake (GZnUp) by 10.4% compared with no application of ZnSB. Among compost types, PMC significantly increased GZnUp by 37.2% over control, which was followed by SMC (34.6%) and FYMC (26.6%). GZnUp showed a positive response toward Zn application. A 49.2% increase over control was recorded in GZnUp with 15 kg Zn $ha^{-1}$, followed by 10 kg Zn $ha^{-1}$ (47.9%) and 5 kg Zn $ha^{-1}$ (40.0%). Higher values of GZnUp (114.37 g $ha^{-1}$) were recorded during 2019–20 than in 2018–19 (101.04 g $ha^{-1}$). The interactive effect of ZnSB × Zn indicated that increasing Zn levels from 0 to 10 kg Zn $ha^{-1}$ increased GZnUp with ZnSB application, while further increases in Zn level did not considerably increase the GZnUp of wheat (Figure 6A). Similarly, the interaction of Zn × C indicated that among different types of compost, SMC and PMC highly increased GZnUp with the application of 5 kg Zn $ha^{-1}$ and slightly increased the values of GZnUp up to 10 kg Zn $ha^{-1}$. Further increases in Zn level did not increase GZnUp (Figure 6B). The interaction of Y × C showed that PMC produced higher values of GZnUp during 2019–20 as compared with 2018–19 (Figure 6C). The interaction of Y × ZnSB × C indicated that the application of PMC enhanced GZnUp with the application of ZnSB during 2018–19 and 2019–20; however, during 2019–20, SMC produced higher values of GZnUp with the application of ZnSB (Figure 6D). The interaction of ZnSB × Zn × C showed that among different types of compost, SMC increased GZnUp with Zn at the rate of 5 kg Zn $ha^{-1}$. Further increases in Zn level did not increase significantly, while in the case of poultry and FYMC, GZnUp increased with each increment of Zn (Figure 7A). Similarly, the interactive effect of Y × ZnSB × C showed that CSM and CPM produced higher values of GZnUp with the application of ZnSB during both years; however, higher values for GZnUp were recorded during 2019–20 (Figure 7B). Similarly, the interactive effect of Y × ZnSB × Zn showed that 10 kg Zn $ha^{-1}$ produced higher values of GZnUp with the application of ZnSB during both years. Further increases in Zn level considerably increased GZnUp (Figure 7C). Similarly, the interactive effect of Y × Zn × C showed that during 2018–19, PMC produced higher values of GZnUp with the application of 15 kg Zn $ha^{-1}$, while during 2019–20, higher values of GZnUp were observed with the application of 10 kg Zn $ha^{-1}$ with the application of PMC (Figure 7D). A significantly positive correlation between straw zinc uptake and shoot Zn concentration, shoot zinc uptake, grain

zinc concentration, grain zinc uptake, straw zinc concentration, and total zinc uptake was observed.

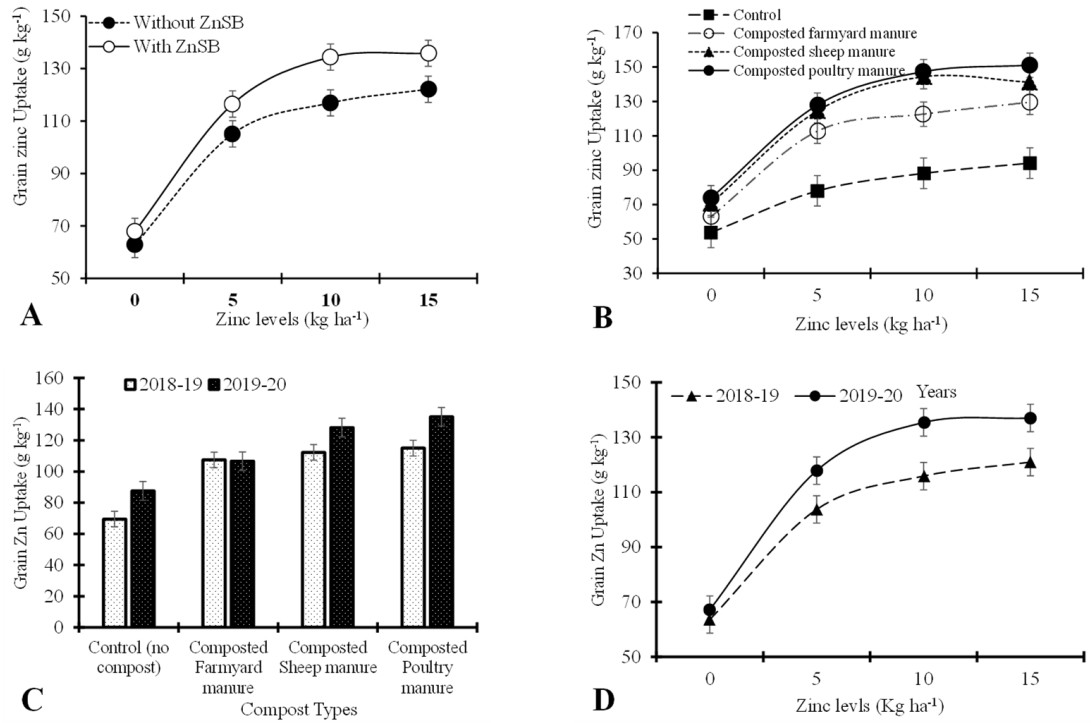

**Figure 6.** Interactive effect of Zn levels and ZnSB (**A**), Zn levels and compost types (**B**), years and compost types (**C**), and years and zinc levels on grain. Zn uptake of wheat (**D**).

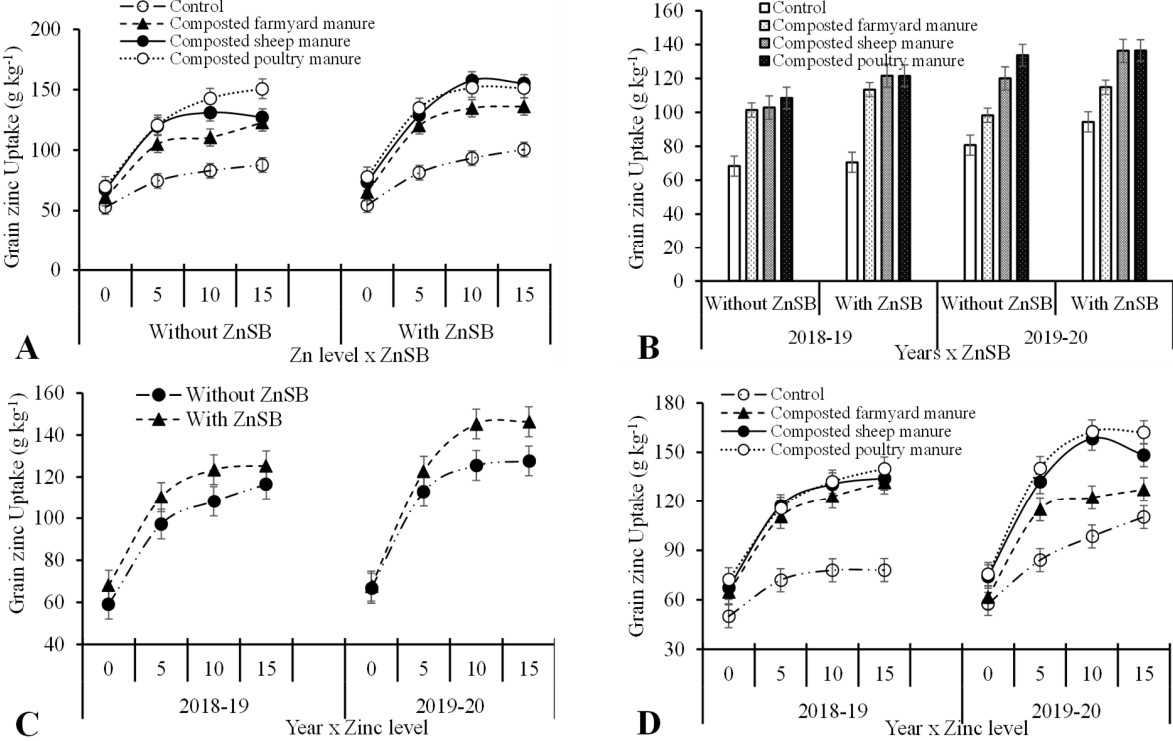

**Figure 7.** Interactive effect of Zn levels, ZnSB and compost types (**A**), years, ZnSB and compost types (**B**), years, Zn levels and ZnSB (**C**), and years, Zn levels and compost types on grain Zn uptake of wheat (**D**).

### 3.5. Straw Zinc Concentrations (mg kg$^{-1}$)

The interactive effects of ZnSB × C, ZnSB × C, Zn × C, Y × ZnSB, Y × C, Y × Zn, Zn × ZnSB × C, Y × ZnSB × C, Y × Zn × C and Y × Zn × ZnSB × C significantly affected straw zinc concentrations (SZnC) (Tables 3 and 4). The application of ZnSB improved SZnC by 9.5% as compared with no application of ZnSB. Among compost types, the application of SMC significantly increased the SZnC of wheat by 24.3%, followed by poultry manure compost (22.8%), over control. In the case of Zn levels, 15 kg Zn ha$^{-1}$ enhanced the SZnC of wheat by 42.1% as compared with control; this was followed by 10 kg Zn ha$^{-1}$ (39%). Higher values of SZnC (21.0 mg kg$^{-1}$) were recorded during 2019–20 than 2018–19 (19.1 mg kg$^{-1}$). The interactive effect of ZnSB × C indicated that ZnSB application with PMC produced higher values of SZnC (Figure 8A). The interaction of ZnSB × Zn indicated that increasing Zn level from 0 to 10 kg Zn ha$^{-1}$ increased SZnC⁻ with ZnSB application, while further increases in Zn level did not considerably increase the values of SZnC of wheat (Figure 8B). The interaction of Zn × C indicated that among different types of compost, SMC and PMC increased values of SZnC with each increment of Zn up to 10 kg Zn ha$^{-1}$. Further increases in Zn level slightly decreased SZnC with the application of SMC (Figure 8C). The interaction of Y × ZnSB indicated that ZnSB produced significantly higher values of SZnC during 2018–19 as compared with 2019–20 (Figure 8D). The interaction of Zn × ZnSB × C indicated that the application of ZnSB produced higher SZnC with the application of SMC and PMC with each increment in Zn up to 10 kg Zn ha$^{-1}$. Further increases in Zn level slightly decreased SZnC with the application of SMC (Figure 9A). Additionally, the interaction of Y × ZnSB × C indicated that the application of PMC enhanced SZnC with the application of ZnSB during 2018–19 and 2019–20; however during 2019–20, SMC produced higher values of SZnC with the application of ZnSB (Figure 9B). Similarly, the interactive effect of Y × Zn × C showed that during 2018–19, PMC produced higher values of SZnC with the application of 15 kg Zn ha$^{-1}$, while during 2019–20, higher values of SZnC were seen with the application of 10 kg Zn ha$^{-1}$ with the application of SMC (Figure 9C). A significantly positive correlation between straw Zn concentrations was observed with shoot Zn concentration, shoot zinc uptake, grain zinc concentration, grain zinc uptake, straw zinc uptake, and total zinc uptake (Table 5).

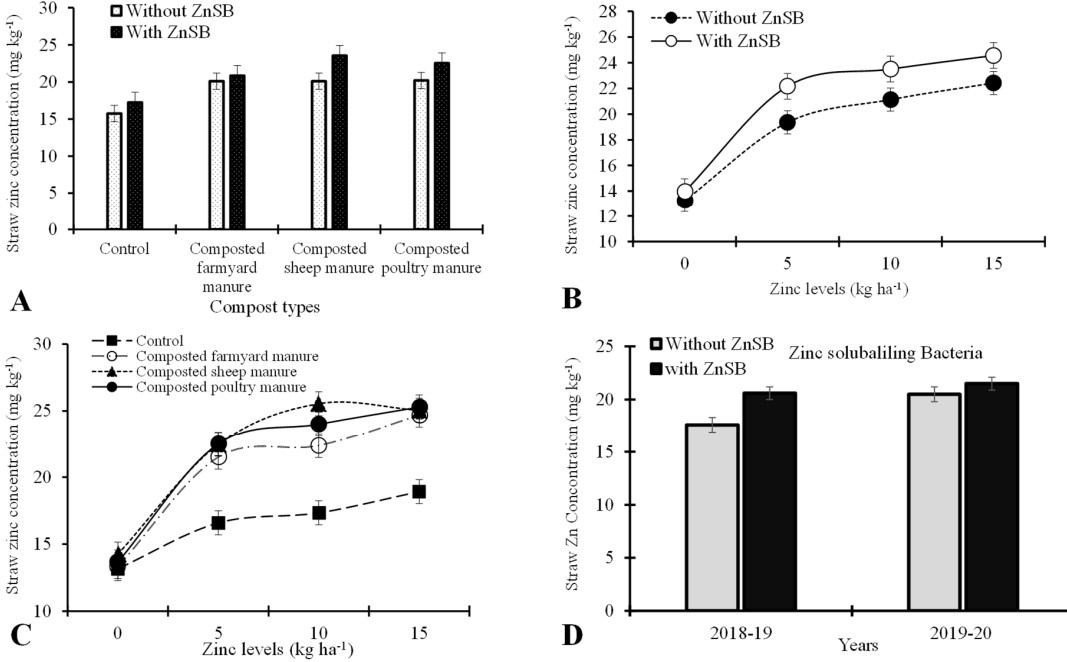

**Figure 8.** Interactive effect of compost types and ZnSB (**A**), Zn levels and ZnSB (**B**), compost types and Zn levels (**C**), and compost types and ZnSB on straw Zn concentration of wheat (**D**).

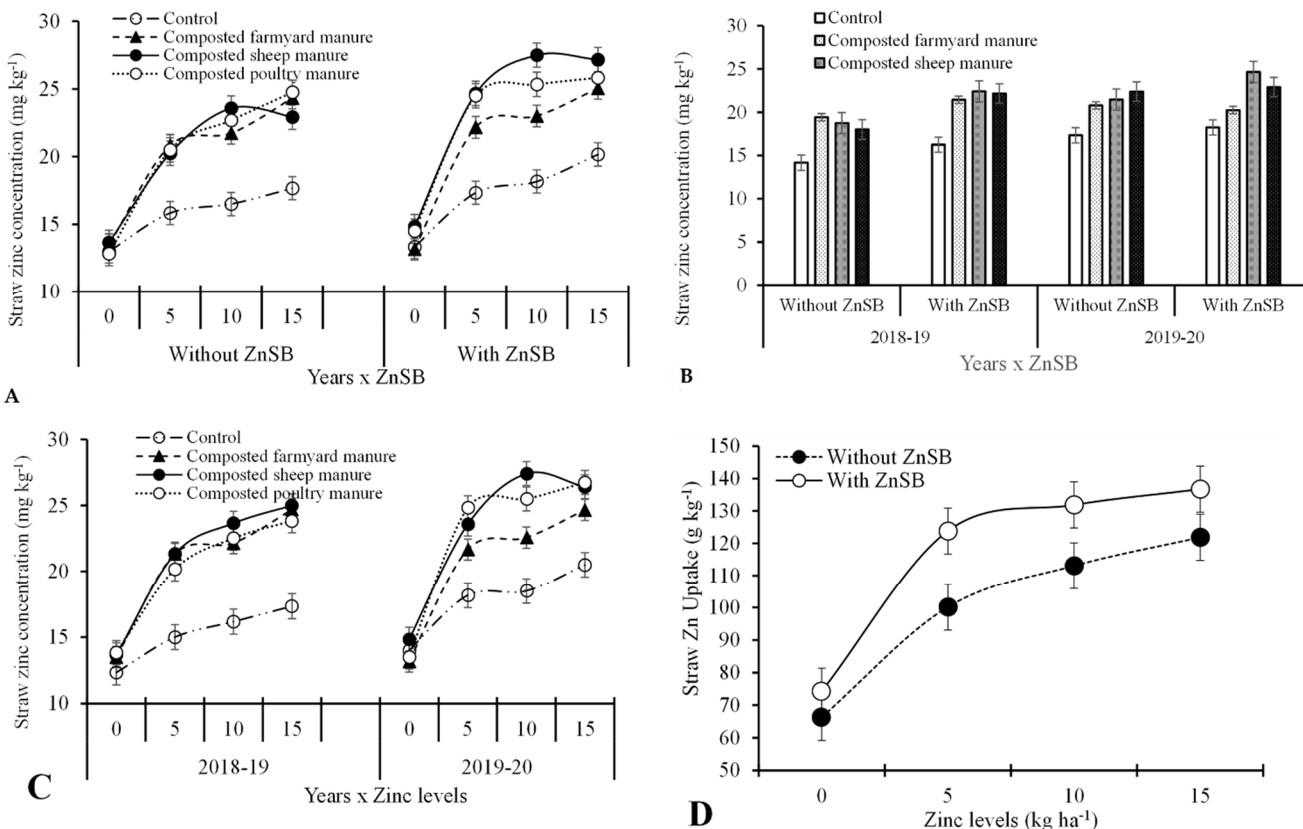

**Figure 9.** Interactive effect of Zn levels, ZnSB and compost types (**A**), years, ZnSB and compost types (**B**), years Zn levels and ZnSB (**C**), and Zn levels and ZnSB on straw Zn uptake of wheat (**D**).

### 3.6. Straw Zinc Uptake (g ha$^{-1}$)

The application of ZnSB enhanced straw zinc uptake (SZnUp) of wheat by 13.9% as compared with no ZnSB application. SZnUp was increased by 35.37% over control with the application of PMC, which was followed by SMC (32.2%) and FYMC (25.3%) (Table 4). The SZnUp of wheat showed a positive response toward Zn application, recording a higher increase in SZnUp of 35.2% over control with the application of 15 kg Zn ha$^{-1}$; this was followed by 10 kg Zn ha$^{-1}$ (32.2%) and 5 kg Zn ha$^{-1}$ (25.3%). The SZnUp of wheat significantly increased by 45.6% compared with control with the application of 15 kg Zn ha$^{-1}$, which was followed by 10 kg Zn ha$^{-1}$ (42.5%) and 5 kg Zn ha$^{-1}$ (37.2%). Higher values of SZnUp (116.17 g ha$^{-1}$) were recorded during 2019–20 than in 2018–19 (100.84 g ha$^{-1}$). The interaction of ZnSB × Zn indicated that increasing Zn levels from 0 to 10 kg Zn ha$^{-1}$ increased SZnUp with the application of ZnSB, while further increases in Zn level did not considerably increase the SZnUp of wheat (Figure 9D). The interaction of Zn × C showed that among different types of compost, PMC increased SZnUp combined with 10 kg Zn ha$^{-1}$, and further increase in Zn level did not increased SZnUp significantly (Figure 10A). The interaction of Y × C showed that PMC produced higher values of SZnUp during 2019–20 as compared with 2018–19 (Figure 10B). The interaction of Y × Zn showed that SZnUp increased with each increment of Zn levels during both years; however, a higher rate of increase was recorded during 2019–20 as compared with 2018–19 (Figure 10C). Application of ZnSB with PMC produced higher values of SZnUp during 2019–20 as compared with 2018–19 (Y × ZnSB × C) (Figure 10D).

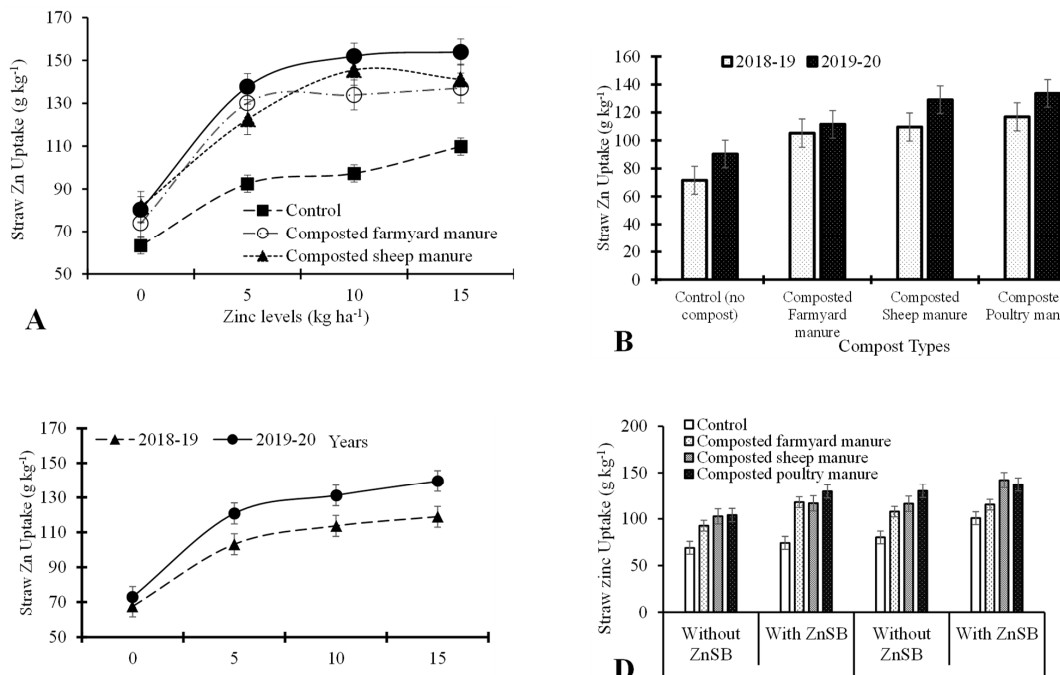

**Figure 10.** Interactive effect of years, compost types and ZnSB (**A**), compost types and ZnSB (**B**), Zn levels and ZnSB (**C**), and Zn levels and ZnSB on straw Zn uptake of wheat (**D**).

### 3.7. Total Zinc Uptake (g ha$^{-1}$)

A maximum increase in total zinc uptake (TZnUp) of 12.2% was recorded with the application of ZnSB as compared with no ZnSB. Among compost types, a 36.2% increase in TZnUp (250.3 g ha$^{-1}$) over control was seen with the application of PMC; this was followed by SMC (33.4%) and FYMC (25.9%) (Table 4). Zn application significantly enhanced TZnUp, which was significantly increased by 47.4% (258.3 g ha$^{-1}$) with the application of 15 kg Zn ha$^{-1}$, followed by 10 kg Zn ha$^{-1}$ (45.3%) and 5 kg Zn ha$^{-1}$ (39.0%). The lowest value of TZnUp was recorded in the control (135.7 g ha$^{-1}$). Higher values of TZnUp (230.5 g ha$^{-1}$) were recorded during 2019–20 than in 2018–19 (201.9 g ha$^{-1}$). The interactive effect of ZnSB × C indicated that the application of ZnSB and compost significantly improved wheat TZnUp, and PMC produced higher values of TZnUp along with the application of ZnSB (Figure 11A). The interaction of ZnSB × Zn indicated that increasing Zn levels from 0 to 10 kg Zn ha$^{-1}$ increased TZnUp along with the application of ZnSB, while further increase in Zn level up to 15 kg Zn ha$^{-1}$ did not increased the TZnUp (Figure 11B). The interaction of Zn × C showed that application of both PMC and SMC linearly increased TZnUp awith an increase in Zn from 0 to 10 kg Zn ha$^{-1}$. Further increases in Zn level up to 15 kg Zn ha$^{-1}$ slightly increased TZnUp combined with PMC, and decreased TZnUp when combined with SMC (Figure 11C). The interaction of Y × C showed that PMC produced higher values of TZnUp during 2019–20 as compared with 2018–19 (Figure 11D). The interaction of Zn × Y indicated that higher TZnUp was recorded during both years of Zn application at the rate of 15 kg Zn ha$^{-1}$; however, during 2019–20, higher values of TZnUp were recorded with the application of Zn at the rate of 15 kg Zn ha$^{-1}$ as compared with 2018–19 (Figure 12A). The interaction of Zn × ZnSB × C indicated that the application of ZnSB produced higher values of TZnUp with the application of SMC with each increment in Zn than without ZnSB application, while SMC and PMC along with ZnSB produced higher values of TZnUp with the application of Zn at the rate of 10 kg Zn ha$^{-1}$ (Figure 12B). The interaction of Y × ZnSB × C indicated that the application of PMC enhanced TZnUp with the application of ZnSB during 2018–19 and 2019–20; however, during 2019–20, SMC produced higher values of GZnUp with the application of ZnSB (Figure 12C). Similarly, the interactive effect of Y × Zn × C showed that during 2018–19, PMC produced higher values of TZnUp with

the application of 15 kg Zn ha$^{-1}$, while during 2019–20, higher values of TZnUp were seen with the application of 10 kg Zn ha$^{-1}$ and PMC (Figure 12D). Table 5 shows that a significantly positive correlation was observed between total zinc uptake and shoot Zn concentration, shoot zinc uptake, grain zinc concentration, grain zinc uptake, straw zinc concentration, and straw zinc uptake.

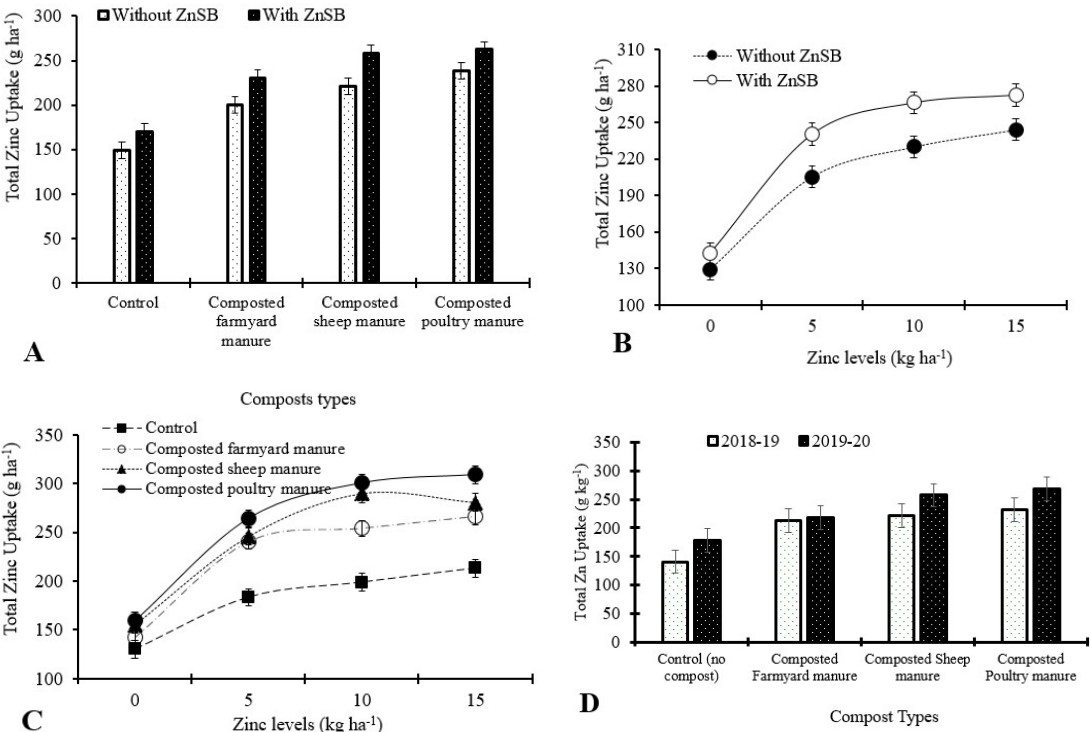

**Figure 11.** Interactive effect of compost types and ZnSB (**A**), Zn levels and ZnSB (**B**), Zn levels and compost types (**C**), and compost types and year on total Zn uptake of wheat (**D**).

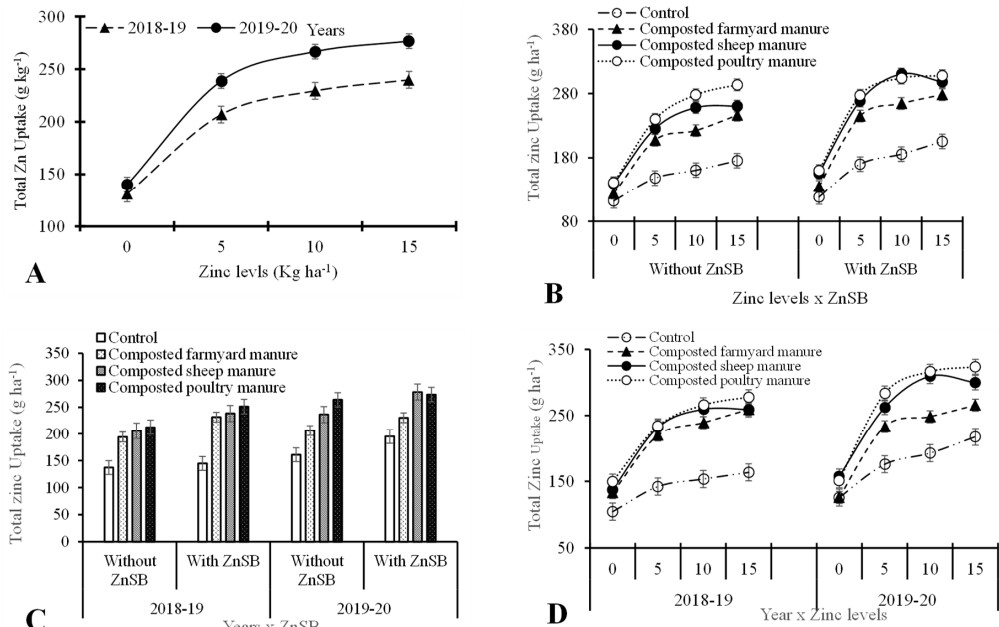

**Figure 12.** Interactive effect of Zn levels and years (**A**), compost types, Zn levels and ZnSB (**B**), years, ZnSB and compost types (**C**), and years, Zn levels and compost types on total Zn uptake of wheat (**D**).

## 4. Discussion

### 4.1. Shoot, Grain, and Straw Zinc Concentrations (mg kg$^{-1}$)

An understanding of nutrient removal by a crop may provide vital information for soil fertility management by comparing the plant's total intake to the total amount of nutrients that are applied from all sources [45]. In the present study, the maximum values of ShZnC, GZnC and SZnC of wheat were recorded with the application of ZnSB as compared to without the application of ZnSB. The higher Zn concentration caused by the usage of ZnSB could be related to the mineralization of Zn in the soil pool. The use of ZnSB lowers the pH of the soil [46,47]; this is presumably due to the generation of organic acids. Furthermore, low rhizospheric pH is linked to H+ extrusion, which produces a significant fall in soil pH [48], resulting in enhanced Zn availability in the soil for plants. Furthermore, by reducing sorption and modifying the properties of soil colloids, ZnSB increases the synthesis of organic acid, which increases the solubilization and eases the mobilization of Zn. Furthermore, the use of ZnSB increased Zn content in whole seeds and seed fractions due to improved Zn uptake owing to better root growth [49], as well as higher ACC deaminase activity [50] and IAA synthesis [51], which may have improved Zn uptake deep from the soil. The use of ZnSB increased the Zn content in wheat shoots, grains, and straws in the current study because ZnSB triggers Zn uptake by altering root shape and boosting sugar and organic acid synthesis in wheat root exudates, which increased Zn uptake in wheat [49].

Higher values of ShZnC, GZnUp and SZnC were recorded during 2019–20 than in 2018–19. Similarly, ZnSB along with compost significantly improved wheat ShZnC, and PMC produced higher ShZnC with ZnSB application. Likewise, PMC increased ShZnC with 15 kg Zn ha$^{-1}$. Similarly, during 2018–19, PMC produced higher values ShZnC during both years with and without ZnSB application; however, during 2019–20, SMC achieved higher values of ShZnC with ZnSB application.

Among compost types, PMC produced the maximum values of ShZnC, GZnUp and SZnC, which was followed by SMC, farmyard manure, and control (no compost application). The higher nutrient uptake with organic manures might be attributed to the solubilization of native nutrients, chelation of complex intermediate organic molecules produced during decomposition of added organic manures, their mobilization and the uptake of different nutrients in different plant parts. The results are in agreement with the findings of Mitra et al. [52] and Sharma et al. [53]. Similarly, Imtiaz et al. [54] reported that compost application significantly affected micronutrient bioavailability. Imtiaz et al. [54], and Marschener [55] showed that compost contributes to Zn uptake through N supply and organic acids, which decrease the soil pH and improve the mobilization of soil Zn in soils. Furthermore, Sujatha and Bhat [56] and Lesnianska et al. [57] stated that compost with lower C:N ratio having a higher content of Zn increase Zn concentrations in the shoot.

The ShZnC, GZnUp and SZnC of wheat showed a positive response toward Zn application. Higher values of the ShZnC, GZnUp and SZnC of wheat were recorded with Zn application in the sequence 15 > 10 > 5 kg Zn ha$^{-1}$, while the lowest values of ShZnC, GZnUp and SZnC were recorded in the control (no Zn application). These results verify the findings of Gao and Grant [58], who also reported that the application of Zn fertilizers, in combination with optimum NPK fertilization, increased grain Zn concentrations. Cakmak [9] and Waters et al. [15] reported that applying Zn enhanced the concentration of Zn in the grain. The enrichment of grain Zn contents is most likely related to Zn feeding, which allows plants to absorb a greater quantity of Zn in leaves, which is subsequently remobilized and allocated in grains [59]. Zn fertilization not only improves nutritional quality but also gradually increases grain production in Zn-deficient soils [9]. Refs. [59,60] found a three-fold increase in grain zinc content. Refs. [15,61] found that increasing zinc supply allowed grains to accumulate large levels of zinc. Ref. [7] also suggested that zinc treatment was preferred since it might boost grain zinc uptake by up to 80%. The application of zinc to wheat grains increased grain zinc absorption according to [62]. According to [3], an increase in grain zinc intake of 10 mg kg$^{-1}$ was adequate to

treat zinc shortage, while Zn application boosted grain zinc. Similarly, Ref. [63] found that applying zinc to grains on a regular basis enhanced grain zinc uptake significantly, as well as seed size and weight.

Higher values of ShZnC, GZnUp and SZnC were recorded during Y2 than Y1. This might be due to the residual and current effects of repeated application of Zn fertilizers, ZnSB and compost, which increased the nutrients' availability to the plant. It might also be due to the equally distributed of rainfall throughout the season and Zn availability, which resulted in better growth of the crop [45]. The PMC produced higher values of ShZnC with the application of ZnSB along with 15 kg Zn ha$^{-1}$. Similar results were reported by Jordão et al. [64], who stated that plants grown in soil amended with compost enriched with Zn showed high Zn concentration in their leaves.

### 4.2. Shoot, Grain and Straw Zinc Uptake (g ha$^{-1}$)

There are a variety of interventions that may be used to combat zinc deficiency, including boosting dietary zinc intake [9,65] and supplementation. It is possible to implement such an intervention by the use of soil bacteria that can mobilize unavailable zinc, improve zinc assimilation, and promote plant development and yield [46,66–68]. There have been several reports of bacteria isolated from soil that have been examined for their mineral zinc-solubilizing activity using a variety of zinc sources [46,69–71]. As a result of their ability to solubilize zinc through the excretion of organic acids, protein extrusion, and the production of chelating agents, ZnSBs can be an excellent supplement to chemical fertilizers. Increased Zn uptake in plant parts can be achieved as a result of the increased Zn uptake in plant parts [39,72,73]. The application of ZnSB significantly increased the values of GZnUp, SZnUp and TZnUp of wheat as compared to without the application of ZnSB. The increased values of GZnUp, SZnUp and TZnUp upon inoculation of different ZnSB strains are mainly due to improved microbial activity, a probable drop in soil pH, and redistribution among native zinc pools resulting in increased zinc availability for crop acquisition [74]. Furthermore, the use of Zn-solubilizing bacteria boosted grain Zn uptake; however, this was accomplished by a variety of mechanisms, including nutrient cycling, transformation, mineralization, and decomposition. Furthermore, the use of ZnSB boosted the Zn uptake of wheat grain, probably as a result of improved plant development and root shape, both of which resulted in increased Zn uptake [68,75,76]. More importantly, low rhizosphere pH as a result of organic acid synthesis is associated with H+ extrusion, which results in a significant reduction in soil pH [48].

When ZnSB was used, it boosted Zn uptake because of increased Zn uptake resulting from improved root growth [49], which was caused by increased activity of ACC deaminase activity [50] and IAA formation [51], which may have resulted in increased Zn uptake deep from the soil. The use of ZnSB increased the uptake of zinc in grain and straw, which activates the uptake of zinc by altering root shape and boosting the formation of sugars and organic acids in wheat root exudates, according to the findings of this study [49].

Among compost types, PMC produced the maximum values of GZnUp (125.06 g ha$^{-1}$. Organic manures have the potential to increase nutrient uptake through a variety of mechanisms, including the solubilization of native nutrients, chelation of complex organic compounds formed during the breakdown of organic manures, their mobilization, and the uptake of different nutrients in different parts of the plant. The results are in agreement with the findings of Mitra et al. [52] and Sharma et al. [53]. Furthermore, Catlett et al. [77] observed that the usage of compost might alter soil characteristics, which may have an impact on Zn concentration in the soil and crop uptake of the element. The nitrogen and organic acids in composts and manures have the potential to lower the pH of the soil and mobilize Zn that is already present in the soil. Poultry manure compost has a low carbon to nitrogen ratio, as well as excellent porosity and water-holding capacity, and it includes the majority of the nutrients in forms that are easily absorbed by plants.

The interaction of Zn × C showed that among different types of compost, SMC increased GZnUp with the application of Zn at the rate of 5 kg Zn ha$^{-1}$. Further increases in

Zn level did not increase significantly, while in the case of PMC and FYMC, GZnUp was increased with each increment of Zn. Similar results were reported by Akinrinde et al. [78], who discovered that the combined application of poultry manure and zinc sulphate improved plant Zn uptake after seeing this phenomenon. Additionally, Math and Trivedi [79] also observed improved yields of wheat and maize, as well as enhanced Zn uptake in both crops, following the application of organic amendments and Zn fertilizers in conjunction with each other.

Higher values of the GZnUp, SznUp and TznUp of wheat were recorded with the application of 15 kg Zn ha$^{-1}$, followed by 10 kg Zn ha$^{-1}$ and 5 kg Zn ha$^{-1}$, while the lowest SZnUp was recorded in control (no Zn application). The increases in GZnUp, SZnUp and TZnUp might be due to the presence of a high amount of Zn in soil solution by the application of a zinc fertilizer, which facilitated greater absorption. Similar results were also reported by Sakal et al. [80], Mollah et al. [81], and Fageria et al. [82], who stated that the application of Zn fertilizers enhanced Zn uptake in plant parts. Similarly, Marschner et al. [83] also reported an increase in GZnUp as a result of Zn application, which involved enzyme activation, maintenance of cellular membrane integrity, and protein synthesis, which are greatly affected by Zn [84–86] and might have increased the GZnUp. The application of Zn application significantly enhanced Zn concentrations in wheat straw in rainfed regimes [87].

Various scientists, including Rafique et al. [88] and Khan et al. [89] stated that the application of Zn at the rate of 5 kg ha$^{-1}$ enhances flag leaf Zn concentrations up to 34 mg kg$^{-1}$. Ref. [90] reported that when Zn fertilizers are used in field circumstances, the Zn content of grain can increase by several orders of magnitude. It has been reported that applying Zn fertilizers or Zn-coated fertilizers to the soil might result in a considerable increase in the Zn content of wheat grain as well as a significant increase in its yield [91]. Additional research conducted by Phattarakul et al. [92] found that the application of Zn can raise the Zn content of whole grains twofold. Foliar application of zinc resulted in an increase in grain zinc concentrations [59,93,94]. According to [14], the reduced efficiency of soil and foliar applications of zinc fertilizer when compared to soil plus foliar applications of zinc fertilizer may be attributed to lower amounts of Zn in wheat shoots when compared to other crops. In one study, Ref. [94] found that foliar 0 $ZnSO_4H_2O$ treatment had the greatest effect on grain Zn, resulting in a 58 percent rise in grain Zn concentration, a 76 percent increase in wheat flour Zn, and a drop in the molar ratio of phytic acid to Zn of up to 50 percent in flour. When Zn is applied in conjunction with foliar spray during the grain development stage, grain Zn content increases by 95 percent, and whole-grain estimated bioavailability increases by 74 percent [95]. Zhang et al. [94] reported that foliar Zn application, either alone or in combination with soil Zn application, led to a considerable improvement in grain Zn concentration, with the concentration increasing from 27.4 mg kg$^{-1}$ to 48.0 ppm as a result of the foliar Zn application. According to [60], zinc treatment increased the uptake of zinc by grains. A favorable relationship has been established between plant zinc uptake and grain zinc concentrations up to 30 mg kg$^{-1}$, as demonstrated by the studies [94,96]. Refs. [59,60] discovered a three-fold rise in the zinc content of grain. Refs. [15,97] found that increasing zinc supply allowed grains to accumulate large levels of zinc. Ref. [35] also suggested that zinc treatment was preferred since it might boost grain zinc uptake by up to 80%. The application of zinc to wheat grains increased grain zinc absorption according to [62]. According to [3], an increase in grain zinc intake of 10 mg kg$^{-1}$ was adequate to treat zinc shortage, while Zn application boosted grain zinc. Similarly, Ref. [98] found that applying zinc to grains on a regular basis enhanced grain zinc uptake significantly, as well as seed size and weight.

## 5. Conclusions

Nutrient management is very important for improving crop productivity and quality in a cereal-based system. Based on our results, agronomic Zn biofortification strategies involving the inclusion of organic and inorganic fertilization and the application of promising zinc solubilizing bacteria, compared to breeding approaches which are long-term, can also

alleviate zinc deficiency and biofortify wheat grain and straw to a greater extent. It was concluded that an integrated approach, i.e., the combined application of fertilizer Zn along with Zn-solubilizing bacteria plus compost, facilitates efficient Zn uptake from the soil, which ultimately produces plants of superior quality. Therefore, it is recommended that the Zn-solubilizing bacteria in combination with organic compost (especially poultry manure compost) and soil-applied Zn (10 kg Zn ha$^{-1}$) is a better choice for farmers to obtain high quality Zn-biofortified wheat crop.

**Author Contributions:** A. designed and supervised the research project and revised the manuscript, and S.K. carried out the lab and field studies, carried out the statistical analysis, wrote the draft paper, and made the figures and tables. I.A. collated and provided the zinc-solubilizing bacteria for this research work. All authors have read and agreed to the published version of the manuscript.

**Funding:** This study was sponsored and supported under the Indigenous PhD Merit Fellowships, by the Higher Education Commission (HEC), Islamabad (No. 518-74570-2AV5-028).

**Institutional Review Board Statement:** Not applicable.

**Informed Consent Statement:** Not applicable.

**Data Availability Statement:** All the data obtained and recorded in this research project will appear online on the website of Higher Education Commission (HEC), Islamabad after the award of PhD degree to the scholar (Shah Khalid).

**Acknowledgments:** The Higher Education Commission (HEC), Islamabad, which sponsored this research project, is highly appreciated.

**Conflicts of Interest:** The authors declare no conflict of interest.

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
