# Peer review of "Enhancing Zinc Biofortification of Wheat through Integration of Zinc, Compost, and Zinc-Solubilizing Bacteria"

_agriculture, doi:10.3390/agriculture12070968_

Round 1

Reviewer 1 Report

The research work carried out by Khalid et al., entitled “Enhancing zinc biofortification of wheat through integration of 2 zinc, compost, and zinc solubilizing bacteria” says that Zinc (Zn) deficiency is a fairly widespread agronomic constraint in many of the world 7 cereal production regions. Zinc is an imperative micronutrient required for optimum plant growth. 8 Low Zn availability in about 50% of the global land resulted in Zn deficiency in cereal grains. A two- 9 year field experiment was conducted at Agronomy Research Farm, The University of Agriculture, 10 Peshawar during Rabi season 2018-19 and 2019-20 to study the impact of Zn levels (0, 5, 10 and 15 11 kg ha-1 ), compost types (control, composted sheep manure (SMC), composted poultry manure 12 (PMC) and farmyard manure compost (FYMC), and Zn solubilizing bacteria (ZnSB) [with (+) and 13 without (-)] on Zn biofortification in order to overcome Zn deficiency. The experiment was set up 14 in a three-replication randomized complete block design. The wheat variety "Pirsabak-2013" was 15 planted at a 30 cm row-to-row spacing. The plot size was kept at 9 cm2 , with 10 rows plot-1 , and the 16 seed was sown at a rate of 100 kg ha-1 . The results showed that ZnSB application increased ShZnC 17 (shoot Zn Conc.) to maximum level of 29.3 mg kg-1 , ShZnUp (shoot Zn Uptake) to 176.0 g ha1 , SZnUp 18 (straw Zn Uptake) to 116.67 g ha-1 , and TZnUp (total Zn uptake) to 230.3 g ha-1 . In the case of com- 19 post types PMC resulted in maximum grain Zn uptake (GZnUp) (28.9 mg kg-1 ), ShZnUp (192.9 g 20 ha1 ), GZnC (33.4 mg kg-1 ), GZnUp (125.06 g ha-1 ), SZnUp (125.26 g ha-1 ), and TZnUp (250.3 g ha-1 ). 21 In case of Zn application, higher ShZnC (31.5 mg kg-1 ), ShZnUp (191.3 g ha1 ), GZnC (34.4 mg kg-1 ), 22 SZnC (23.5 mg kg-1 ), GZnUp (128.98 g ha-1 ), SZnUp (129.29 g ha-1 ), TZnUp (258.3 g ha-1 ) was calcu- 23 lated with the use of 15 kg Zn ha-1 which was either statistically similar or followed by 10 kg Zn ha- 24 1 . A strong positive correlation was found among uptake by different plant parts (ZnG, ZnS, 25 ShZnUp, GZnUp, SZnUp, TZnUp). It was concluded that combined application of PMC and Zn at 26 the rate of 10 kg Zn ha-1 along with ZnSB (+) improved Zn biofortification and uptake in wheat crop 27 under Zn deficient soils.

The authors have done good work, still, I recommend revision:

My comments are the following: -

Line 29: There is a huge number of keywords, and a few of them are irrelevant. Please try to keep 5-6 keywords only.

Line 164: All the formulae must be written properly by using the equation tool in the word or some other.

Conclusion must be rewritten as currently it has mainly results. 

Author Response

First Reviewer

S.No.

Comments/Suggestions of the Honorable Reviewer

Replies by the Author

1

The authors have done good work, still, I recommend revision:

Thank you very much for encouraging comments. Yes we have revised the manuscript further and improved a lot.

2

Line 29: There is a huge number of keywords, and a few of them are irrelevant. Please try to keep 5-6 keywords only.

Appreciated, the Keywords reduced to five words please (incorporated in line no. 29)

3

Line 164: All the formulae must be written properly by using the equation tool in the word or some other.

Appreciated, all formulas are converted through the equation tool as per suggestions (incorporated in line no. 166 to 168)

4

The conclusion must be rewritten as currently, it has mainly results. 

Appreciated, revised the conclusion as per the suggestion  (line 561 to 573)

Reviewer 2 Report

The paper describes a large number of ways of getting more Zinc into a wheat crop. The units mg/kg and g/kg get mixed up a lot in the current text. The paper is of interest to wheat farmers. The manuscript is a bit dense with figures and tables, but this is okay. The relative performance of each type of fertilizer is about the same with respect to the control. My conclusion would be that there are diverse successful approaches for biofortification and choosing a fertilizer should be based on local convenience.

The English could be improved. The data shows that several techniques are equally valid for Zinc biofortification, the conclusion picks one of each category,

Author Response

Second Reviewer

S. No.

Comments/Suggestions of the Honorable Reviewer

Replies by the Author

1

The paper describes a large number of ways of getting more Zinc into a wheat crop

Thank you so much. It is actually a unique research work for the award of PhD degree. We tried our best to get maximum beneficial effect.

2

The units mg/kg and g/kg get mixed up a lot in the current text

Appreciated, Corrected accordingly line 203,399,

3

The paper is of interest to wheat farmers. The manuscript is a bit dense with figures and tables, but this is okay.

Actually it is a PhD research work for the thesis. Different interactions were studied to get maximum results. We appreciate your comments.

4

The relative performance of each type of fertilizer is about the same with respect to the control.

Yes, different treatments showed better results than control. However, poultry manure compost shows a higher relative performance in comparison with other organic composts.

5

My conclusion would be that there are diverse successful approaches for biofortification and choosing a fertilizer should be based on local convenience.

Appreciated, conclusion revised according to the comments of both reviewers. Page 561 to 573

6

The English could be improved.

We checked the manuscript twice and corrected minor errors in English, thanks

7

The data shows that several techniques are equally valid for Zinc biofortification, the conclusion picks one of each category,

Yes it is true. All the fertilized treatments showed a significant increase over control. Conclusion revised Page 561 to 573
